# ERK Inhibition Increases RANKL-Induced Osteoclast Differentiation in RAW 264.7 Cells by Stimulating AMPK Activation and RANK Expression and Inhibiting Anti-Osteoclastogenic Factor Expression

**DOI:** 10.3390/ijms232113512

**Published:** 2022-11-04

**Authors:** Eun-Bi Choi, Taiwo Samuel Agidigbi, In-Soon Kang, Chaekyun Kim

**Affiliations:** 1Laboratory for Leukocyte Signaling Research, Department of Pharmacology and Toxicology, College of Medicine, Inha University, Incheon 22212, Korea; 2BK21 Program in Biomedical Science & Engineering, Inha University, Incheon 22212, Korea; 3Convergent Research Center for Metabolism and Immunoregulation, Inha University, Incheon 22212, Korea

**Keywords:** osteoclast, macrophage, ERK, RANKL, AMPK, anti-osteoclastogenic mediators

## Abstract

Bone absorption is necessary for the maintenance of bone homeostasis. An osteoclast (OC) is a monocyte–macrophage lineage cell that absorbs bone tissue. Extracellular signal-regulated kinases (ERKs) are known to play important roles in regulating OC growth and differentiation. In this study, we examined specific downstream signal pathways affected by ERK inhibition during OC differentiation. Our results showed that the ERK inhibitors PD98059 and U0126 increased receptor activator of NF-κB ligand (RANKL)-induced OC differentiation in RAW 264.7 cells, implying a negative role in OC differentiation. This is supported by the effect of ERK2-specific small interfering RNA on increasing OC differentiation. In contrast to our findings regarding the RAW 264.7 cells, the ERK inhibitors attenuated the differentiation of bone marrow-derived cells into OCs. The ERK inhibitors significantly increased the phosphorylation of adenosine 5′-monophosphate-activated protein kinase (AMPK) but not the activation of p38 MAPK, Lyn, and mTOR. In addition, while the ERK inhibition increased the expression of the RANKL receptor RANK, it decreased the expression of negative mediators of OC differentiation, such as interferon regulatory factor-8, B-cell lymphoma 6, and interferon-γ. These dichotomous effects of ERK inhibition suggest that while ERKs may play positive roles in bone marrow-derived cells, ERKs may also play negative regulatory roles in RAW 264.7 cells. These data provide important information for drug development utilizing ERK inhibitors in OC-related disease treatment.

## 1. Introduction

Bone homeostasis is a tightly regulated process mediated by various bone cells including osteoblasts, osteocytes, and osteoclasts (OCs) [1,2,3,4]. Osteoblasts and OCs function in tandem to remodel or reshape bone structures through bone resorption mediated by OCs and bone formation by osteoblasts. OCs are present on bone surfaces at the sites of active bone resorption, participating in the dissolution of mineralized tissues [5,6,7]. OCs are large multinucleated cells derived from the monocyte–macrophage lineage [8]. The differentiation and activity of OCs are controlled by two essential cytokines: macrophage colony-stimulating factor (M-CSF) and receptor activator of nuclear factor-κB ligand (RANKL) [9,10]. M-CSF primarily promotes the proliferation and survival of OC precursors, whereas RANKL stimulates the differentiation of OC precursors into mature OCs [11].

M-CSF and RANKL play critical roles in OC differentiation by activating mitogen-activated protein kinases (MAPKs) [12]. Interaction of M-CSF with its receptor c-Fms activates the MEK/ERK pathway and regulates the proliferation and survival of OC precursors [13]. Interaction of RANKL with RANK recruits TRAF6 and triggers extracellular signal-regulated kinase (ERK) activation, which is implicated for its roles in OC differentiation [14,15]. The activation of MAPKs by M-CSF or RANKL differs with respect to the extent, duration, and isoform specificity of MAPKs, hence determining the fates of OC proliferation and differentiation [14]. M-CSF primarily promotes cellular proliferation by activating ERK, whereas RANKL induces a switch to differentiation. ERK regulates diverse cellular functions, including survival, proliferation, apoptosis, polarity, and differentiation in OCs [16,17,18,19]. RANKL/RANK to ERK signaling cascades positively regulate the formation and function of OCs [7,19,20,21]. However, previous studies have shown that ERK negatively regulates OC differentiation [16,22,23].

Bone metabolism is closely linked to energy homeostasis. RANKL induces adenosine 5′-monophosphate-activated protein kinase (AMPK) activation in bone marrow-derived monocytes/macrophages (BMMs) and stimulates OC formation. AMPK, a heterotrimeric complex that consists of catalytic α and non-catalytic βγ subunits, is ubiquitously present in various tissues. AMPK acts as an “energy sensor” that regulates cellular energy homeostasis to activate glucose and fatty acid uptake and oxidation when cellular energy levels are low [24]. AMPK plays a negative role in RANKL-induced OC formation in BMMs [25,26]. Deletion of AMPK increases osteoclastogenesis and reduces bone mass [27,28,29], whereas activation of AMPK suppresses OC differentiation [30]. Moreover, ERK and AMPK pathways play contradictory roles in OC differentiation [31,32].

RANKL/RANK signaling downregulates the expression of negative regulators of osteoclastogenesis, such as interferon regulatory factor-8 (IRF8), B-cell lymphoma 6 (Bcl6), and interferon-γ (IFN-γ) [33,34,35,36]. These molecules inhibit osteoclastogenesis by targeting the nuclear factor of activated T cells 1 (NFATc1). The expression of IRF8 in OC precursors is downregulated during the initial phase of RANKL-induced OC differentiation, and IRF8-deficient mice exhibit severe osteoporosis owing to the increased number of OCs [33]. Overexpression of Bcl6 inhibits osteoclastogenesis, and Bcl6-deficient mice show increased OC differentiation, leading to severe osteoporosis [34]. Contradictory results have been obtained upon examination of the role of IFN-γ in OC differentiation. IFN-γ inhibits OC formation and bone resorption [36,37,38]. However, other studies have shown that IFN-γ enhances OC formation and promotes bone resorption [39,40].

In our previous study, ERK played a negative role in RANKL-induced OC differentiation of RAW 264.7 cells by increasing the levels of reduced glutathione (GSH) and antioxidant genes [23]. However, the precise function and mechanism of ERK in OC differentiation remain unclear. In the present study, we attempted to further characterize the role of ERK in OC differentiation by examining the molecules that regulate OCs. ERK inhibition using ERK inhibitors or ERK2-specific small interfering RNA (siRNA) increased OC differentiation by increasing AMPK activation and RANK expression and inhibiting the expression of negative mediators of OC differentiation in RAW 264.7 cells.

## 2. Results

### 2.1. ERK Inhibition Stimulated RANKL-Induced OC Differentiation in RAW 264.7 Cells

RAW 264.7 cells are macrophage-like cells that express RANK, and they are frequently used as precursors for OC differentiation. Therefore, we used RAW 264.7 cells as OC precursors in the present study. Consistent with our previous findings [23], the selective MEK/ERK inhibitors PD98059 and U0126 increased TRAP-positive OC numbers in RAW 264.7 cells (Figure 1A,B), while inhibiting ERK phosphorylation (Figure 1D,E). Moreover, ERK inhibition increased tartrate-resistant acid phosphatase (TRAP) activity and F-actin formation (Figure 1C,F). The number of OCs containing fewer than 10 nuclei was similar between the control and PD98059-treated groups. However, an increase in the number of OCs containing more than 10 nuclei was observed in the PD98059-treated groups (Figure 1G,H), indicating that ERK inhibition increases the size and number of OCs. PD98059 also promotes the differentiation of skeletal muscle and myoblasts [41,42]. Therefore, our data suggest that ERK negatively regulates RANKL-induced OC differentiation of RAW 264.7 cells. This proposal is to be further substantiated by the results of ERK2 siRNA transfection experiments. 

### 2.2. ERK Inhibition Suppressed RANKL-Induced OC Formation in BMMs

Although RAW 264.7 cells are frequently used as OC precursors, they behave differently from BMMs. M-CSF is a prerequisite for the differentiation of BMMs; however, RAW 264.7 cells readily differentiate into OCs in the absence of M-CSF. Therefore, we treated murine BMMs with PD98059 and U0126 in the presence of M-CSF and RANKL and differentiated them into OCs. Compared with RAW 264.7 cells, PD98059 and U0126 inhibited OC formation in BMMs in a dose-dependent manner (Figure 2). These results suggest that ERK inhibition suppresses M-CSF and RANKL signaling during OC differentiation in BMMs, which leads to defects in cell proliferation and differentiation. This is supported by the results that treatment of ERK inhibitors inhibited cell growth and OC differentiation of BMMs (Appendix A).

### 2.3. ERK Knockdown Increased RANKL-Induced OC Differentiation in RAW 264.7 Cells

To demonstrate the negative role of ERK in RAW 264.7-derived OCs, we knocked down ERK in RAW 264.7 cells using ERK2 siRNA and differentiated the cells into OCs. Both ERK1 and ERK2 play essential roles in supporting osteoclastogenesis. In this study, we focused on the effects of ERK2 on molecular signaling mechanisms activated by RANKL because of following reasons: First, ERK2 is expressed more dominantly than ERK1 in most mammalian tissues [43,44]. Secondly, ERK2 has been relatively less thoroughly studied compared to ERK1 in OCs [18]. ERK expression was reduced in ERK2 siRNA-transfected cells compared to that in scRNA-transfected cells (Figure 3A–C). However, the number of OCs in the ERK2 siRNA-transfected group was higher than that in scRNA-transfected cells (Figure 3D,E). The OCs derived from ERK2 siRNA-transfected cells were significantly larger than those derived from scRNA-transfected cells (Figure 3F,G). Moreover, ERK2 siRNA transfection markedly increased the expression of OC markers, including dendritic cell-specific transmembrane protein (DC-STAMP) and cathepsin K, and the master transcription factor of OC, NFATc1 (Figure 3H–J). These results strongly suggest that ERK negatively regulates RANKL-induced OC differentiation in RAW 264.7 cells. 

### 2.4. ERK Inhibition Increased RANK Expression

RANK is a member of the tumor necrosis factor receptor family that binds only to RANKL. It is expressed in immune cells and OCs [10,45]. Its expression is not a typical feature of hematopoietic progenitors but is acquired in specific lineages during hematopoiesis [46]. Here, we determined whether ERK inhibition could alter RANK expression. RANK expression was increased in PD98059- or U0126-treated cells and ERK2 siRNA-transfected cells (Figure 4), indicating that ERK negatively regulates RANK expression in OCs. However, this result does not clearly indicate whether increased RANK expression enhanced OC differentiation or whether OC differentiation increased RANK expression.

### 2.5. ERK Inhibition Increased AMPK Phosphorylation

OC differentiation employs many kinases, such as MAPKs, Src family kinases, PI3K/Akt, Btk/Tec, and AMPK [14,47,48,49]. We determined whether ERK regulated OC differentiation by modulating these kinases, including p38 MAPK, Lyn, and AMPK. PD98059 had no effect on the phosphorylation of Lyn and mTOR (Figure 5A–C). There is evidence that p38 MAPK positively regulates OC differentiation and that p38 MAPK activation stimulates osteoclastogenesis via the inhibition of ERK-mediated OC precursor proliferation [16,17]. Phosphorylation of p38 MAPK was slightly increased by PD98059; however, the difference was not statistically significant. AMPK phosphorylation was significantly increased by PD98059 and in ERK2 siRNA-transfected cells (Figure 5D–F), suggesting an opposing role of ERK and AMPK in osteoclastogenesis. 

### 2.6. ERK Inhibition Suppressed the Expression of Negative Mediators of OC Differentiation

In our previous study, PD98059 increased NFATc1 expression in RAW 264.7-derived OCs [23]. Transcriptional repressors of NFATc1 inhibit OC differentiation [33,34,50]. To investigate whether ERK could modulate the expression of negative mediators of OC differentiation, we analyzed the corresponding transcript levels using quantitative real-time polymerase chain reaction (qRT-PCR). As shown in Figure 6A–F, mRNA expression of IRF8, Bcl6, and IFN-γ was decreased by ERK inhibitors and in ERK2 siRNA-transfected cells. Moreover, Bcl6 protein expression was decreased upon ERK suppression (Figure 6G–I). IRF8 and Bcl6 bind to NFATc1 and suppress NFATc1 autoamplification and the expression of NFATc1-targeted marker genes of OC [50]. These results suggest that ERK suppression increases OC differentiation by decreasing the expression of the negative regulators of osteoclastogenesis.

## 3. Discussion

We have previously showed that ERK negatively regulates RANKL-induced OC differentiation in RAW 264.7 cells by increasing cellular GSH levels and nuclear factor erythroid 2-related factor 2 expression [23]. However, our findings did not sufficiently elucidate the role of ERK in OC differentiation. In this study, we investigated the effects of ERK on signal pathways involved in OC differentiation. BMMs can be differentiated into OCs in vitro with M-CSF followed by treatment with a combination of M-CSF and RANKL, whereas RAW 264.7 cells can be easily differentiated only in the presence of RANKL. BMMs and RAW 264.7 cells were efficiently differentiated into OCs, confirming that both the M-CSF and RANKL signaling pathways were involved in the differentiation of BMMs, whereas only the RANKL pathway was involved in OC differentiation of RAW 264.7 cells. However, the role of M-CSF and RANKL in OC differentiation does not seem straightforward. In this regard, BMMs and RAW 264.7 cells differ in the conditions required for their differentiation into OCs. It has been reported that in the presence of M-CSF, RANKL stimulated cell proliferation of BMMs at 0–48 h but inhibited their proliferation at 48–96 h [51]. This suggests that OC differentiation requires different signal pathways in a stage-dependent manner. 

Many studies have been undertaken using RAW 264.7 cells because RAW 264.7 cell-derived OCs closely resembled BMM-derived OCs with regard to morphology, time to differentiate OC precursors into OCs, and OC-forming potential. Thus, it has been believed that OCs derived from RAW 264.7 cells are similar to those derived from BMMs. However, Ng et al. [52] compared the characteristics of OCs from BMMs and RAW 264.7 cells using quantitative proteomics, and showed a low concordance between BMMs and RAW 264.7 cells (R^2^ ≈ 0.13). Furthermore, RAW 264.7 cells demonstrated constitutive activation of ERK and Akt, differing from BMMs in the required signal activation for OC differentiation [52], which suggests discrepancies between BMMs and RAW 264.7 cells in OC differentiation.

To the best of our knowledge, ERK stimulates OC differentiation in BMMs [19,20,21,53], which contradicts our findings regarding RAW 264.7 cells. Blockade of ERK stimulated the differentiation of RAW 264.7 cells into OCs but inhibited the differentiation of BMMs (Figure 1 and Figure 2). Such distinctive results between the two cell types suggest that ERK may play a binary role during OC differentiation, switching OC differentiation stages. We believe that the alteration of ERK roles during OC differentiation is regulated by M-CSF and RANKL signaling, at least in vitro. Since M-CSF-induced ERK activation primarily promotes the proliferation and survival of OC precursors [13], the inhibition of ERK in BMMs should suppress the M-CSF-induced proliferation of cells. However, this positive ERK effect may occur at the early stage of osteoclastogenesis. This is supported by the results that PD98059 inhibited M-CSF-induced cell growth at an early stage of BMM differentiation (Appendix A). However, RAW 264.7 cells can differentiate independent of M-CSF, bypassing the early stages of OC differentiation. So, the sole effects of ERK were seen in the RAW 264.7 cells at the late stages of RANKL-induced OC differentiation. Similar phenomena were observed in the positive effect of DNA synthesis on osteoclastogenesis at an early proliferative phase but inversely at latter stages [51]. Therefore, RANKL-induced OC differentiation may be coupled with the antiproliferative activity of RANKL, supporting our view that OC differentiation from RAW 264.7 cells and BMMs uses the same signaling pathways acting in potentially opposite roles. In this study, we demonstrate that while ERK activation is a positive regulator of BMMs differentiating into OCs, blockade of ERK show an opposite effect on RAW 264.7 cells rather promoting them to differentiate into OCs. However, a direct correlation between two cell types was not possible because the optimal cell densities in our culture systems were not the same.

AMPK is composed of an αβγ heterotrimer with multiple subunit isoforms, and most AMPK subunits are expressed in bone cells [27]. The main subunit is the α1 subunit, and the presence of the α2 subunit is negligible in bone cells. The β1 and β2 subunits are similarly expressed. The γ1 subunit is the major γ isoform in bone cells; however, γ2 and γ3 are not expressed [27,54,55]. Nonetheless, conflicting results have been reported on the role of AMPK in OC differentiation and function. RANKL induces AMPKα activation and OC formation in BMMs. The AMPK activator 5-aminoimidazole-4-carboxamide ribonucleotide (AICAR, acadesine) increases the number of OCs and stimulates bone loss and bone resorption [27]. However, inhibition of AMPK with compound C (dorsomorphin) or AMPKα1 siRNA also increases OC formation and bone resorption in BMMs, and AICAR and metformin inhibit OC differentiation [25,26]. Moreover, AMPK plays a negative role in osteoprotegerin-mediated OC inhibition [30]. Contrasting roles of AMPK and ERK in OCs have been reported in metformin-treated BMMs. Metformin and glycyrrhizin inhibit OC differentiation by inhibiting ERK and activating AMPK [31,32]. Therefore, we suggest that the negative effect of ERK during differentiation of RAW 264.7 cells into OCs is mediated by modulating AMPK signaling. However, the key molecules or signaling pathways that switch the role of ERK during OC differentiation have not been clarified, and further studies are needed to elucidate these factors in the future. In addition, the binary interplay between MAPK and AMPK signaling has been studied in cancer biology [56].

In the present study, inhibition of ERK increased OC differentiation in RAW 264.7 cells but decreased it in BMMs. This suggests multiple roles of ERK in osteoclastogenesis. In the early period of our culture system with BMMs, ERK signals play a positive role in osteoclastogenesis, as many other researchers have previously demonstrated. In contrast, our results clearly indicate that ERK signaling shows suppressive roles in RANKL-induced RAW 264.7 cell OC differentiation. This suppressive effect of ERK on RAW 264.7 cell OC differentiation is possibly due to the antiproliferative activity of RANKL. Furthermore, ERK inhibited AMPK activation and RANK expression in RAW 264.7-derived OCs and stimulated the expression of negative mediators of OC differentiation, such as IRF8, Bcl6, and IFN-γ (Figure 7). Thus, we propose that ERK signaling may be a target for the treatment of diseases that show abnormal bone resorption, such as osteoporosis, rheumatoid arthritis, and osteopetrosis. However, precaution should be taken when ERK signaling is targeted for drug development to treat OC-related diseases.

## 4. Materials and Methods

### 4.1. Reagents and Antibodies

Minimum essential medium (α-MEM), fetal bovine serum (FBS), phosphate-buffered saline (PBS), penicillin, and streptomycin were purchased from HyClone (Logan, UT, USA). Recombinant murine M-CSF and human RANKL were purchased from PeproTech (Rocky Hill, NJ, USA). Oligonucleotides were purchased from Bioneer (Daejeon, Korea). Alexa Flour 555-phalloidin, enhanced chemiluminescent solution (ECL), and a bicinchoninic acid kit were purchased from Thermo Scientific (Rockford, IL, USA). Antibodies against ERK, phospho-ERK, p38 MAPK, phospho-p38 MAPK, AMPKα, phospho-AMPKα, and Bcl6 were purchased from Cell Signaling Technology (Danvers, MA, USA). Antibodies against Lyn, phospho-Lyn, mTOR, and phospho-mTOR were purchased from Santa Cruz Biotechnology (Santa Cruz, CA, USA). All other chemicals were purchased from Sigma-Aldrich (St. Louis, MO, USA) unless stated otherwise.

### 4.2. Differentiation of RAW 264.7 Cells into OCs

RAW 264.7 cells (ATCC, Manassas, VA, USA) were plated in 96-well plates (4 × 10^3^/well) and incubated in α-MEM supplemented with 10% FBS, 100 U/mL penicillin, and 100 mg/mL streptomycin in the presence of 50 ng/mL RANKL at 37 °C. The cells were treated with selective potent inhibitors of MEK/ERK, namely PD98059 (0.5–10 μM) and U0126 (0.5–2 μM), for 4 days during OC differentiation. PD98059 and U0126 were dissolved in dimethyl sulfoxide (DMSO) and the final concentration of DMSO was 0.1%. The culture medium was replaced on the third day of differentiation, and OC formation was assessed by counting TRAP-positive cells with more than three nuclei. OC area was measured using ImageJ (1.51v) software (National Institutes of Health, Bethesda, MD, USA).

### 4.3. Differentiation of BMMs into OCs

C57BL/6J mice (Jackson Laboratory, Bar Harbor, VT, USA) were housed under specific pathogen-free conditions at the animal facility of Inha University. All procedures were conducted in accordance with the institutional guidelines approved by the Animal Care and Use Committee of Inha University (INHA-200107-680-1). Bone marrow cells were isolated from 6-to-8-week-old male mice and cultured overnight in complete α-MEM containing 10 ng/mL M-CSF. Subsequently, non-adherent cells were collected the next day and continuously cultured with 30 ng/mL M-CSF for 3 days. The cells (2 × 10^4^ cells/96-well plate or 5 × 10^4^ cells/48-well plate) were then cultured for an additional 4 days with a combination of 30 ng/mL M-CSF and 50 ng/mL RANKL to induce OC differentiation. During this incubation period, the cells were treated with ERK inhibitors. 

### 4.4. Transfection of ERK2-Specific siRNA

RAW 264.7 cells were transfected with ERK2-specific siRNA and non-targeting scrambled RNA (scRNA). RAW 264.7 cells were incubated with 200 nM siRNA in Opti-MEM (Gibco, Carlsbad, CA, USA) containing Lipofectamine RNAi-MAX (Invitrogen, Carlsbad, CA, USA) for 6 h at 37 °C. The medium was then replaced with fresh complete α-MEM containing 50 ng/mL RANKL followed by culturing for 4 days for OC differentiation.

### 4.5. TRAP Staining

TRAP-positive cells were determined using a leukocyte acid phosphate assay kit (Sigma) as previously described [57]. Briefly, the cells were washed with PBS (pH 7.0–7.2), dried for 1 h at 20 ± 5 °C, and fixed with a fixing solution containing 65% acetone, 25% citrate solution, and 8% formaldehyde. The fixed cells were then incubated with the TRAP staining solution for 1 h at 37 °C in the dark. After washing twice with water, the cells were counterstained with haematoxylin III for 2 min and washed with water. TRAP-positive cells that contained three or more nuclei were considered mature OCs when visualized under an Axioplan 2 microscope (Zeiss, Jena, Germany).

### 4.6. RNA Preparation and qRT-PCR

Total RNA was extracted from OCs using TRIzol reagent (Invitrogen) and then reverse-transcribed according to the manufacturer’s protocol (Takara Bio, Shuzo, Japan). Quantitative PCR was performed using an RT-PCR detection system (Bio-Rad CFX 96, Berkeley, CA, USA) with SYBR Green PCR Master Mix (Toyobo, Osaka, Japan), and the primers are listed in Table 1. Relative gene expression was normalized to that of glyceraldehyde 3-phosphate dehydrogenase (GAPDH) based on changes in the threshold cycle (Ct).

### 4.7. Western Blot Analysis

Cell lysates were prepared as previously described [58]. Briefly, the total protein lysate was mixed with a protein loading dye and subjected to 10% SDS-PAGE. The separated proteins were then transferred onto a polyvinylidene fluoride membrane (Millipore, Bedford, MA, USA) and incubated with specific primary antibodies followed by an appropriate secondary antibody. Signals were developed using an ECL kit. Integrated densitometry was performed to determine the intensity of the scanned films using ImageJ.

### 4.8. TRAP Activity

TRAP activity was determined as previously described [23]. Briefly, cells were incubated with 10 mM sodium tartrate and 5 mM p-nitrophenyl phosphate in 50 mM citrate buffer (pH 4.5) for 30 min at 37 °C. The enzyme reaction was then terminated with 0.1N NaOH. Absorbance was measured at 405 nm using a VersaMax microplate reader (Molecular Devices) equipped with SoftMax software.

### 4.9. Actin Ring Formation Assay

Mature OCs on sterile cover glass were fixed with 3.7% formaldehyde for 10 min and permeabilized with 0.1% Triton X-100 for 5 min. F-actin and nuclei were stained with 0.66 μM Alexa Flour 555-phalloidin for 30 min and 1 μg/mL 4′, 6-diamidino-2-phenylindole (DAPI) for 5 min in the dark at 20 ± 5 °C. Cells were washed with PBS twice and observed under a fluorescence microscope (Olympus, Tokyo, Japan). 

### 4.10. Statistical Analysis

A two-tailed Student’s *t*-test (paired) was performed for comparison within groups using Microsoft Excel software, and one-way analysis of variance (ANOVA) was performed for comparisons between groups using Prism software (version 9.0; GraphPad, La Jolla, CA, USA). Data are expressed as the mean ± SD of more than three independent experiments. A *p*-value less than 0.05 is considered statistically significant. 

## Figures and Tables

**Figure 1 ijms-23-13512-f001:**
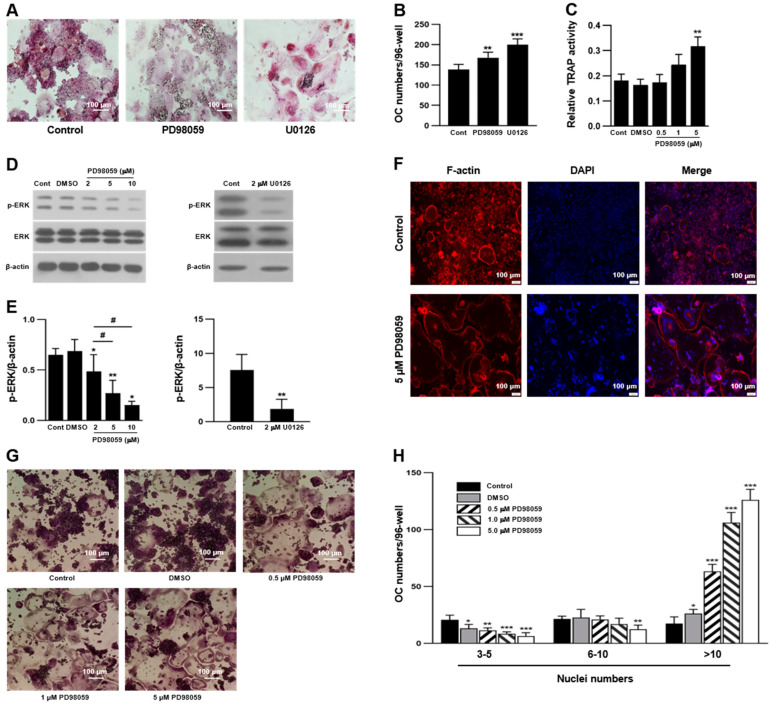
Inhibition of extracellular signal-regulated kinase (ERK) increased osteoclast (OC) differentiation in RAW 264.7 cells. Cells were treated with PD98059 and U0126 in the presence of 50 ng/mL receptor activator of NF-κB ligand (RANKL) for 4 days. (**A**,**B**) Effect of ERK inhibitors on the formation of tartrate-resistant acid phosphatase (TRAP)-positive multinucleated OCs (n = 4 independent experiments, each performed in triplicate). (**C**) Effect of PD98059 on TRAP activity of OCs (n = 3). (**D**,**E**) Effect of PD98059 and U0126 on ERK phosphorylation of RAW 264.7-derived OCs (n = 3). (**F**) Actin ring formation of OCs, which were stained by Alexa 555-phalloidin (F-actin) and DAPI (nuclei), respectively (n = 3). (**G**) Dose effect of PD98059 on OC differentiation (n = 3 independent experiments, each performed in triplicate). (**H**) Number of nuclei per OC (n = 3). Each result represents the mean ± SD. * *p* < 0.05, ** *p* < 0.01, and *** *p* < 0.001 vs. control (**B**,**C**,**E**,**H**) or dimethyl sulfoxide (DMSO) (**E**), and ^#^
*p* < 0.05 vs. 2 μM PD98059 (**E**).

**Figure 2 ijms-23-13512-f002:**
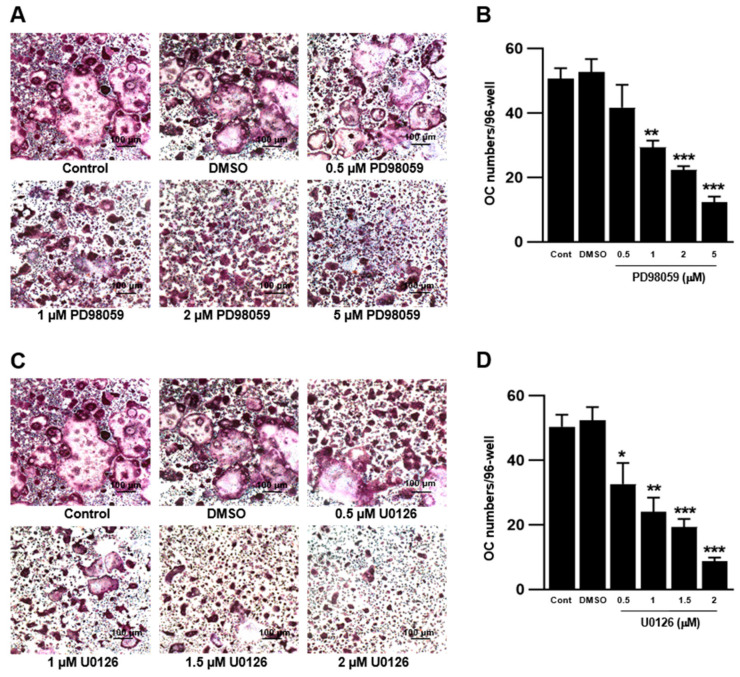
Inhibition of ERK decreased OC differentiation in bone marrow monocytes (BMMs). Cells were cultured with 30 ng/mL macrophage colony-stimulating factor (M-CSF) for 3 days and then treated with PD98059 and U0126 in the presence of 30 ng/mL M-CSF and 50 ng/mL RANKL for 4 days. (**A**,**B**) Effect of PD98059 on OC differentiation from BMMs (n = 3 independent experiments, each performed in triplicate). (**C**,**D**) Effect of U0126 on OC differentiation from BMMs (n = 3 independent experiments, each performed in triplicate). Each result represents the mean ± SD. * *p* < 0.05, ** *p* < 0.01, and *** *p* < 0.001 vs. DMSO.

**Figure 3 ijms-23-13512-f003:**
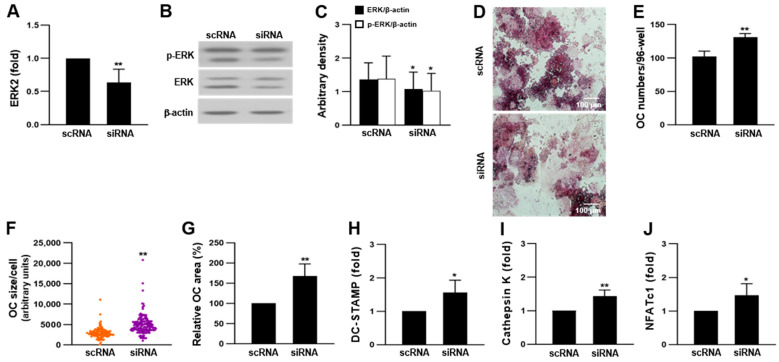
Downregulation of ERK with ERK2 siRNA increased OC differentiation in RAW 264.7 cells. Cells were transfected with ERK2 siRNA and differentiated into OCs in the presence of 50 ng/mL RANKL for 4 days. (**A**–**C**) Effect of ERK2 siRNA on the expression of ERK (n = 5 for mRNA and n = 3 for protein). (**D**,**E**) Effect of ERK2 siRNA on OC differentiation (n = 3 independent experiments, each performed in triplicate). (**F**,**G**) Relative size of OC and OC area per well of OCs of ERK2 siRNA-transfected cells (n = 3). (**H**–**J**) Quantitative analyses of mRNA expression of OC markers, DC-STAMP and cathepsin K, and NFATc1 was measured by qRT-PCR (n = 3). Each result represents the mean ± SD. * *p* < 0.05 and ** *p* < 0.01 vs. scRNA.

**Figure 4 ijms-23-13512-f004:**
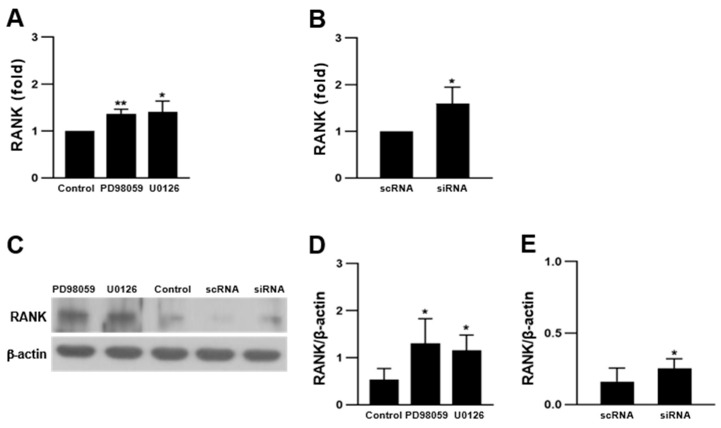
Inhibition of ERK increased expression of RANK. RAW 264.7 cells were treated with 5 μM PD98059 and 2 μM U0126, or were transfected with ERK2 siRNA, and differentiated into OCs in the presence of 50 ng/mL RANKL for 4 days. (**A**,**B**) Quantitative analyses of mRNA expression by ERK inhibitors and ERK2 siRNA (n = 3). (**C**–**E**) Protein expression of RANK in ERK inhibitor-treated and ERK2 siRNA-transfected cells (n = 3). Each result represents the mean ± SD. * *p* < 0.05 and ** *p* < 0.01 vs. control or scRNA.

**Figure 5 ijms-23-13512-f005:**
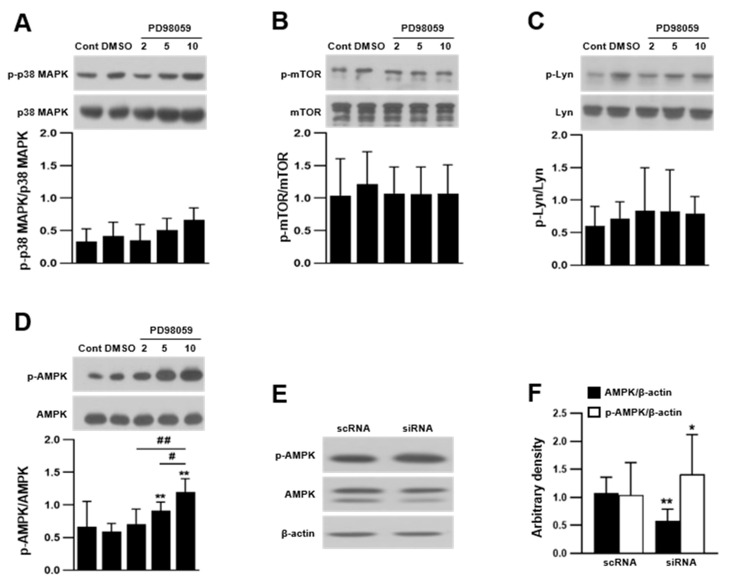
Inhibition of ERK increased adenosine 5′-monophosphate-activated protein kinase (AMPK) phosphorylation. Cell lysates were prepared from PD98059-treated OCs and ERK2 siRNA-transfected OCs, and activation of kinases was measured by Western blotting. Phosphorylation of p38 MAPK (n = 4) (**A**), mTOR (n = 3) (**B**), Lyn (n = 3) (**C**), and AMPK (n = 6) (**D**). (**E**,**F**) AMPK phosphorylation was determined in OCs of ERK2 siRNA-transfected cells (n = 5). Each result represents the mean ± SD. * *p* < 0.05 and ** *p* < 0.01 vs. control or scRNA, and ^#^
*p* < 0.05 and ^##^
*p* < 0.01.

**Figure 6 ijms-23-13512-f006:**
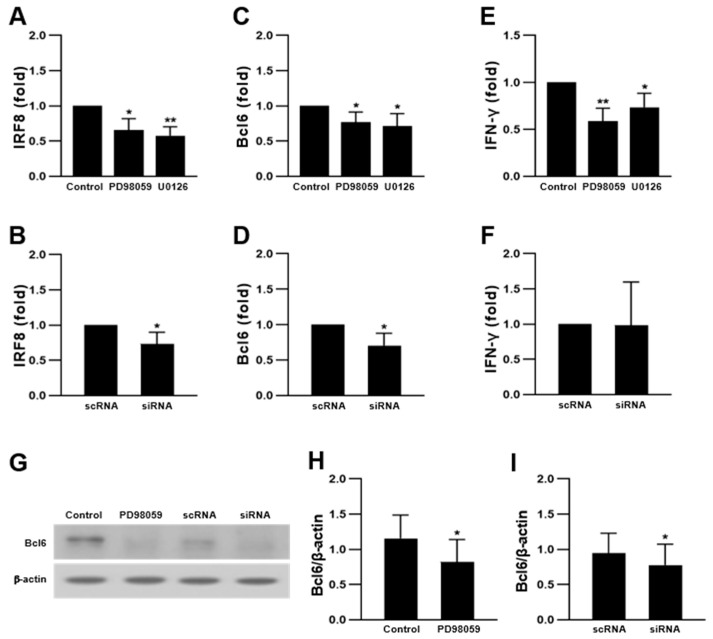
Inhibition of ERK suppressed the expression of inhibitory mediators of OC differentiation. RAW 264.7 cells treated with 5 μM PD98059 and 2 μM U0126 or transfected with ERK2 siRNA, and differentiated into OCs in the presence of 50 ng/mL RANKL for 4 days. Quantitative analyses of mRNA expression of IRF8 (**A**,**B**), Bcl6 (**C**,**D**), and IFN-γ (**E**,**F**) (n = 3). (**G**–**I**) Protein expression of Bcl6 in PD98059-treated and ERK2 siRNA-transfected cells (n = 3). Each result represents the mean ± SD. * *p* < 0.05 and ** *p* < 0.01 vs. control or scRNA.

**Figure 7 ijms-23-13512-f007:**
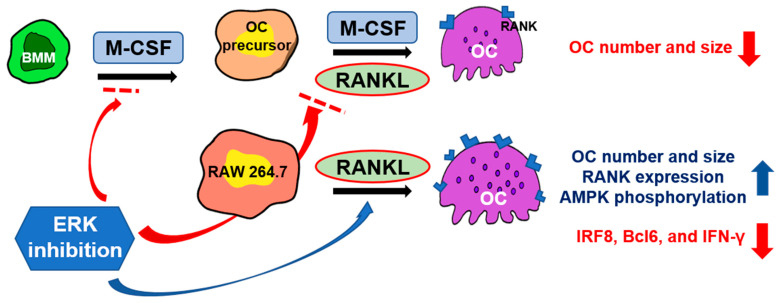
Proposed molecular mechanism that ERK regulates OC differentiation. Inhibition of ERK increased OC differentiation in RAW 264.7 cells by increasing RANK expression and AMPK phosphorylation and inhibiting the expression of negative mediators of osteoclastogenesis, such as IRF8, Bcl6, and IFN-γ.

**Table 1 ijms-23-13512-t001:** List of qRT-PCR primer sequences.

Gene	Forward Primer	Reverse Primer
*Bcl6*	CAG AGA TGT GCC TCC ATA CTG C	CTC CTC AGA GAA ACG GCA GTC A
*Cathepsin K*	GAA GAA GAC TCA CCA GAA GCA G	TCC AGG TTA TGG GCA GAG ATT
*DC-STAMP*	TC CTC CAT GAA CAA ACA GTT CCA A	AG ACG TGG TTT AGG AAT GCA GCT C
*ERK2*	TCT GCA CCG TGA CCT CAA	GCC AGG CCA AAG TCA CAG
*IFN-* *γ*	CAG CAA CAG CAA GGC GAA AAA GG	TTT CCG CTT CCT TGA GGC TGG AT
*IRF8*	CAA TCA GGA GGR GGA TGC TTC C	GTT CAG AGC ACA GCG RAA CCT C
*NFATc1*	GGG TCA GTG TGA CCG AAG AT	GGA AGT CAG AAG TGG GTG GA
*RANK*	CTA ATC CAG GGA AGC AAA T	GAC ACG GGC ATA GAG TCA GTT C
*GAPDH*	CCT TCC GTG TTC CTA CCC C	CCC AAG ATG CCC TTC AGT

## Data Availability

The data presented in this study are included in the article and are also available on request from the corresponding author.

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
