# Peer review of "ERK Inhibition Increases RANKL-Induced Osteoclast Differentiation in RAW 264.7 Cells by Stimulating AMPK Activation and RANK Expression and Inhibiting Anti-Osteoclastogenic Factor Expression"

_ijms, 2022, doi:10.3390/ijms232113512_

Round 1
Reviewer 1 Report
The author should provide the structures of PD98059 and U0126.
This study described inhibition of ERK increased OC differentiation in RAW 264.7 253 cells. Further, ERK inhibited AMPK activation and 257 RANK expression in RAW 264.7-derived OCs, and stimulated the expression of negative 258 mediators of OC differentiation, such as IRF8, Bcl6, and IFN-a. So, ERK may be examined as a target for the treatment of diseases that show abnor-260 mal bone resorption, such as osteoporosis, rheumatoid arthritis, and osteopetrosis.
This manuscript is written well and has some innovative.
However, the author should provide the structures of PD98059 and U0126 before publication on IJMS.
Author Response
Dear Reviewer,
We appreciate the careful review and constructive suggestions. Please find enclosed the revised version of the manuscript IJMS-1880789 entitled “Inhibition of ERK increases RANKL-induced osteoclast differentiation in RAW 264.7 cells by stimulating AMPK activation and RANK expression but inhibiting the expressions of anti-osteoclastogenic factor”.
- The author should provide the structures of PD98059 and U0126 before publication on IJMS.
R: Thank you for your kind review and comments. The structures of PD98059 and U0126 are as follow. We have not included them in the text because their structures are well known.
Please see the attached file for the structure.

Reviewer 2 Report
In the work “Inhibition of ERK Increases RANKL-induced Osteoclast Differentiation in RAW 264.7 Cells by Stimulating AMPK Activation and RANK Expression but Inhibiting the Expressions of Anti-Osteoclastogenic Factor” Eun Bi Choi and coworkers study in deep the role of ERK signaling in osteoclast differentiation by using well-characterized in vitro models of osteoclastogenesis.
The study is the natural consequence of the previous article of the group, i.e., Agidigbi TS, et al., Free Radic Res. 2020, and for this reason, it is very poorly original. Indeed, the effects of the ERK inhibitor PD98059 on OC differentiation have been published by the authors. The addition of another inhibitor (U0126) or the use of ERK2 silencing in the current study doesn’t improve the originality of the work and the levels of knowledge about the proven role of the ERK in OC biology. For example, the investigation of the role of ERK in OC activity (by analyzing TRAP activity) could increase the originality of the work.
Moreover, starting from the abstract, which is poorly organized, the manuscript is not well written and extensive editing of the language is needed. The text widely is not careful in the description of the results, legends, and figures leading to confusing conclusions, that are not fully supported by the results.
Major concerns:
Abstract:
The abstract should be organized in clear and structured sections. It is confusing.
Introduction:
The introduction section is poorly organized and the information is fragmentary. Essential information and relative references regardless of the role of ERK in OC differentiation (actually in the discussion section) should be reported in this section to guide the readers to the results.
Results:
In general, during the description of the results, a lot of sentences should be moved to the discussion section. Moreover, the authors should report the number of performed experiments in each legend.
Figure 1:
1) The authors should include the effects of ERK inhibitors on OC activity (i.e. TRAP activity) to increase the originality of the work (panels A and E)
2) Why the authors did not show the effects of U0126 in ERK phosphorylation (panel C)
3) the authors should report in panel F only the results showing the significant effects of different doses of PD98059 in OC with more than 10 nuclei in a single graph, and they should report the no significant effects in a supplement figure. It might be most clear. Why they don’t report the effects of the U0126?
The immunofluorescence staining for F-actin by phalloidin might be a simple and elegant method to examine and count multinucleated cells with F-actin rings, which are also indispensable for OC activity (Wang, J et al., Front. Pharmacol. 2019, 10, 1188).
Figure 2:
1) the authors should explain clearly because they investigate the effect of ERK inhibition in BMMs by rephasing the 101-108 lines.
2) in lines 108-110 the authors suggest and effects of the ERK inhibition on BMMs proliferation to sustain their results. Where is the data? it is necessary
3) what happens if the inhibition of ERK occurs from the start of BMMs treatment with M-CSF and not only after 3 days? This is a necessary experiment to sustain the conclusions of the authors
Figure 3:
1) the authors should explain because they investigated only the silencing for ERK2 excluding ERK1.
2) panel B is related to panel A? the silencing of a gene must be demonstrated at RNA levels.
Figure 5:
1) in lines 166-168 the authors said “Phosphorylation of p38 MAPK was slightly increased by 166 PD98059; however, it did not reach statistical significance”. In the images reported in panel A, the increase of p38 is none evident.
Discussion
1) A lot of evidence reported the positive role of ERK in RANKL-dependent osteogenesis in the same experimental conditions of the authors (Refs 44-52 in Lee K et al., Int. J. Mol. Sci. 2018). The authors should explain this discrepancy.
2) The authors said, “Therefore, both M-207 CSF and RANKL signaling pathways were involved in the differentiation of BMMs, 208 whereas RAW 264.7 cells used only RANKL pathway for OC differentiation” (line 207-209). The authors should consider the article of Zhang H, et al., 2021 (doi:10.3389/fendo.2021.642676) in which osteoclasts differentiation was induced by incubation of RAW264.7 cells in the presence of RANKL and M-CSF. The use of the ERK inhibitors in this model might clarify and demonstrate the double role of ERK suggested by the authors during OC differentiation.
Author Response
Dear Reviewer,
We appreciate the careful review and constructive suggestions. Please find enclosed the revised version of the manuscript IJMS-1880789 entitled “Inhibition of ERK increases RANKL-induced osteoclast differentiation in RAW 264.7 cells by stimulating AMPK activation and RANK expression but inhibiting the expressions of anti-osteoclastogenic factor”.

Reviewer 3 Report
The report by Choi et al. demonstrates that ERK negatively regulates RANKL-induced osteoclast differentiation of RAW264.7 cells and investigates the molecular mechanism of ERK-specific inhibition that leads to osteoclast differentiation. The manuscript is well structured and the cited references are relevant. However, the experimental desing needs to be improved in order to better support the author's hypothesis that ERK inhibition at later stages of OC development increases OC differentiation. ERK inhibition by U0126 and PD98059 have an opposite effect in BMM cells and supresses their OC differentiation. The authors have treated BMM cells with ERK inhibitors at earlier stage of their OC differentiation. The experiment has to be completed by treatment of BMM cells at a later stage of OC differentiation, as indicated by fig.7 - during the stimulation with M-CSF and RANKL, which leads to maturation of BMM-derived OC precursors to OCs. It is important to investigate this and to check whether ERK inhibition at this later stage will increase OC differentiation of BMM cells, which can be considered as a better model for OC differentiation studies than the tumor-derived RAW264.7 cells.
Other points that need to be addressed are:
- Figures 1, 2 and 3 contain graphs that show "OC numbers/96 well". Please, clarify if the data represent OC numbers per 96-well plate.
- Figure 3 E : How did the authors measured the size of OCs? It is not indicated in Materials and methods section. Please explain "OC area/cell (arbitrary units)". Please indicate percentage of OC area per well.
- Figure 4 C: RANK expression in ERK2 siRNA-transfected cells is not increased compared to the control. Revise subsection 2.4., L147-148. How do the authors explain the extremely low expresion of RANK in scRNA-transfected cells?
- Subsection 2.5: Explain how the results presented in Fig. 5 support the statement that MAPK and AMPK signaling reversibly regulate OC differentiation. The authors show a trend for increased p-p38 MAPK following PD98059 treatment.
- Revise the Discussion section. Comparison of BMM- and RAW264.7-derived OCs needs to be expanded, as well as the discussion on the opposite effects of ERK inhibition in BMM cells and RAW264.7.
Author Response

(The authors gave the same response as above.)

Round 2
Reviewer 2 Report
The authors highlighted or included missed formal aspects along with the text in the current version of the manuscript as required by the Reviewer i.e. re-organization of the abstract, results, and discussion sections, partially improving the quality of their work. I appreciate the effort of the authors to answer all points raised by the reviewer and to perform a part of missed experiments. However, they did not perform essential required analyses to sustain their conclusions, i.e. proliferation assay, the effects of U0126 in ERK phosphorylation (if this treatment is not particularly relevant the relative results should be moved to supplementary material), and the treatment with ERK inhibitors or ERK2 silencing at different time points. The authors based the discussion of their results on the different effects between RANKL and MCF in OC proliferation, without showing the effects of ERK inhibition in this parameter in both cells type. I strongly believe in the quality of our results, but I think that the data are still too preliminary to achieve correct conclusions and for publication in a journal with a high impact. The ongoing experiments of the authors should be included in this manuscript to give relevance to the work in the field of bone remodeling.
Minor points to raise:
1) Abstract is still confusing in the description of methods and results
2) Figure 1H. From which analyses did the count derive? It is not clear.
3) The justification for the use of ERK2 silencing should be included in the manuscript
4) Having finished the samples cannot be a justification for not performing the analyses required by a Reviewer.
Author Response
October 20, 2022
Dear Reviewer,
We appreciate your careful review and constructive suggestions on our manuscript. Your comments have helped a lot in improving our manuscript. We have made the appropriate corrections and explanation point-by-point as suggested by you, and changes made in the manuscript are marked using track changes. Below, we give our detailed response to your comments. We hope this improved version of manuscript, together with our answers below, will adequately address all of the questions and contribute to a clearer presentation of the data.
Thank you very much for considering our manuscript for publication in IJMS.
Sincerely,
Chaekyun Kim, Ph. D.
Laboratory for Leukocyte Signaling Research,
Department of Pharmacology,
Inha University School of Medicine,
100 Inha-ro, Michuhol-gu, Incheon 22212, Korea
E-mail: chaekyun@inha.ac.kr
Phone: 82-32-860-9874, Fax: 82-32-885-8302
Response to Reviewers Comments
The authors highlighted or included missed formal aspects along with the text in the current version of the manuscript as required by the Reviewer i.e. re-organization of the abstract, results, and discussion sections, partially improving the quality of their work. I appreciate the effort of the authors to answer all points raised by the reviewer and to perform a part of missed experiments. However, they did not perform essential required analyses to sustain their conclusions, i.e. proliferation assay, the effects of U0126 in ERK phosphorylation (if this treatment is not particularly relevant the relative results should be moved to supplementary material), and the treatment with ERK inhibitors or ERK2 silencing at different time points. The authors based the discussion of their results on the different effects between RANKL and MCF in OC proliferation, without showing the effects of ERK inhibition in this parameter in both cells type. I strongly believe in the quality of our results, but I think that the data are still too preliminary to achieve correct conclusions and for publication in a journal with a high impact. The ongoing experiments of the authors should be included in this manuscript to give relevance to the work in the field of bone remodeling.
R: Thank you for your kind review and comments. In this revision, we have included additional data, such as proliferation assay (Fig. S1), the effects of U0126 in ERK phosphorylation (Fig. 1D & E), and the treatment with ERK inhibitors at different time points of BMMs differentiation (Fig. S2), and the mRNA expression of ERK siRNA-transfected OCs (Fig. 3A).
- Minor points to raise:
1) Abstract is still confusing in the description of methods and results
R: The abstract has been modified to be clearer. We hope this modified version of abstract will adequately organized to a clearer meaning.
2) Figure 1H. From which analyses did the count derive? It is not clear.
R: We counted the number of nuclei in OC after TRAP staining under the microscope.
3) The justification for the use of ERK2 silencing should be included in the manuscript
R: We have added the following sentences in the Result section 2.3.
Both ERK1 and ERK2 play essential roles in supporting osteoclastogenesis. In this study, we focused on the effects of ERK2 on molecular signaling mechanisms activated by RANKL because of following reasons. First, ERK2 is expressed more dominantly than ERK1 in most mammalian tissues [43,44]. Secondly, ERK2 has been relatively less thoroughly studied compared to ERK1 in OCs [18].
4) Having finished the samples cannot be a justification for not performing the analyses required by a Reviewer.
R: We agree with you. We have added ERK2 RNA expression result in Figure 3A.

Reviewer 3 Report
The manuscript has been significantly improved. However, there are parts in it that still need to be revised.
Regerding my major concern about the experiment with BMMs the authors have corrected the protocol for treatment with ERK inhibitors stating that they wanted to compare BMMs and RAW264.7 cells at the same differentiation state. This is a good point but how do they explain the conclusions drawn in the end of the manuscript (L711, L891-892). If you compare cells at the same differentiation stage why do you present conclusions for early and late stage of differentiation? Please explain and revise the conclusions.
Figures 1,2,3: Replace "OC numbers/96-well" with "OC numbers/96-well plate".
L537-538: the text added at the end of the sentence needs to be clarified. It can be used in a separate sentence. An exapmle: During this incubation period the cells were treated with ERK inhibitors.
Author Response
October 20, 2022
Dear Reviewer,
We appreciate your careful review and constructive suggestions on our manuscript. Your comments have helped a lot in improving our manuscript. We have made the appropriate corrections and explanation point-by-point as suggested by you, and changes made in the manuscript are marked using track changes. Below, we give our detailed response to your comments. We hope this improved version of manuscript, together with our answers below, will adequately address all of the questions and contribute to a clearer presentation of the data.
Thank you very much for considering our manuscript for publication in IJMS.
Sincerely,
Chaekyun Kim, Ph. D.
Laboratory for Leukocyte Signaling Research,
Department of Pharmacology,
Inha University School of Medicine,
100 Inha-ro, Michuhol-gu, Incheon 22212, Korea
E-mail: chaekyun@inha.ac.kr
Phone: 82-32-860-9874, Fax: 82-32-885-8302
Response to Reviewers Comments
The manuscript has been significantly improved. However, there are parts in it that still need to be revised. Regerding my major concern about the experiment with BMMs the authors have corrected the protocol for treatment with ERK inhibitors stating that they wanted to compare BMMs and RAW264.7 cells at the same differentiation state. This is a good point but how do they explain the conclusions drawn in the end of the manuscript (L711, L891-892). If you compare cells at the same differentiation stage why do you present conclusions for early and late stage of differentiation? Please explain and revise the conclusions.
We have tried to explain and revise the manuscript, in particular abstract and conclusion. We hope this modified version of manuscript will adequately answer your questions and comments.
- Figures 1,2,3: Replace "OC numbers/96-well" with "OC numbers/96-well plate".
R: It represents the number of cells in a 96-well, so I think 96-well is more suitable than 96-well plate.
- L537-538: the text added at the end of the sentence needs to be clarified. It can be used in a separate sentence. An exapmle: During this incubation period the cells were treated with ERK inhibitors.
R: Thank you. We have changer the sentence as you recommended.